# OpenReview forum: "LLMS LOST IN TRANSLATION: M-ALERT UNCOVERS CROSS-LINGUISTIC SAFETY GAPS"
_ICLR.cc/2025/Workshop/BuildingTrust — BuildingTrust_

### Official Review · Reviewer_Yagk · 2025-02-17
**Nice step towards multi-lingual safety evaluations with minor quality concerns and questions regarding some conclusions.**

**Rating:** 7
**Confidence:** 4

**Review:**

This work presents a new multilingual dataset of 75k toxic prompts from 5 languages which are organized in a taxonomy.
15k prompts are taken from the english-only ALERT dataset, and 60k of the prompts are translated automatically.
29 LLMs from several families are benchmarked on this dataset.
The authors claim to show "significant safety inconsistencies across languages and topics"


Pro:
- multilingual safety is underexplored but important!
- fine-grained structure and evaluation is good to give nuanced view on safety
- comprehensive evaluation of many models
- clear description of the limitations of the dataset & judge model that was used
- clear description of the experimental setup (I feel confident that I could reproduce the experiments in the paper)

Cons:
- translations done by models, only 0.066% of  translations are checked by human experts
- 7% of translations (in some languages up to 9%) are wrong according to human evaluation, limiting the significance of inter-lingual differences. The highest inter-language gap for a single model is 6.8%, which is less than the estimated transcription error rate. Thus we urge the authors to soften their claims regarding the significance of inter-lingual differences.
- discrepancies between crime_propaganda results in M-ALERT (en) and ALERT.
- already appears almost saturated, with models achieving close to 100%, except on certain ambiguous categories such as cannabis.


Questions:

Can you explain why no model achieves more than 73% on english crime_propaganda? On the original ALERT all models were able to achieve >90%, sometimes up to 100%.

In Table 2, the errors for MetricX are huge - why is that?

---

### Official Review · Reviewer_9tBk · 2025-02-24
**Summary**

**Rating:** 6
**Confidence:** 4

**Review:**

### Summary of the Paper:

This paper introduces a new multilingual benchmark called M-ALERT. It is based on the ALERT taxonomy and focuses on five languages: English, French, German, Italian, and Spanish. The authors provide a safety evaluation using this dataset on ten large language models (LLMs), highlighting differences in safety across languages.

### Strengths:

The authors evaluate translation using two independent metrics: COMET and MetricX which confirm decent translation results across languages

They provide a comprehensive evaluation of their methods across different model sizes and families.

### Weaknesses:

The manual assessment for a subset of 100 random prompts could be extended to a larger sample to improve stability.

The assessment focuses on five widely available languages, but the analysis lacks at least one language with lower availability, such as a Slavic languages.

### Questions and Suggestions for Improvement:
1. Why didn’t you filter out samples with poor translation metrics?

2. The translation metrics in Table 2 could be presented more clearly, as they have similar values but different ranges. The MetricX follows a "lower is better" principle, while the COMET follows the opposite.

3. How exactly is safety scoring conducted? In the Overall Safety Discrepancies section, safety is discussed in relation to time. This should be clarified—what time frame do you mean? How did you determine the safety threshold (0-90-99%)?

4. The analysis of base models could be less emphasized, as the finding that safety scores are higher for instruction-tuned models compared to base models is an expected and well-known result.

---

### Official Review · Reviewer_EHLB · 2025-03-01
**Practical multilingual benchmark and insights**

**Rating:** 7
**Confidence:** 4

**Review:**

This paper introduces M-ALERT, a multilingual benchmark designed to evaluate safety in large language models (LLMs) across five languages: English, French, German, Italian, and Spanish. The authors build upon the existing ALERT taxonomy to comprehensively assess LLM safety, emphasizing language-specific vulnerabilities and inconsistencies across different categories. The authors plan to publicly released the dataset, enhancing transparency and facilitating further research in the field.

Strengths:
1. Comprehensive and Novel Benchmark: The paper addresses a critical gap by extending the ALERT benchmark to multiple languages and publicly releasing the dataset, thereby significantly contributing to the robustness, transparency, and generalizability of LLM safety evaluations.
2. In-depth Experimental Analysis: Experiments across multiple state-of-the-art LLMs reveal meaningful insights, particularly highlighting cases where safety performance diverges notably between languages. Such detailed scrutiny, including language-specific and category-specific analyses, greatly enhances the value of their findings.
3. Inter-language Consistency Metric to identify inter-language disparities


Weaknesses:

1. Potential Evaluator Bias: Relying primarily on LlamaGuard-3 for safety scoring introduces potential biases, particularly if this evaluator is not equally proficient across all languages and contexts evaluated.
2. Translation Quality Challenges: Although robustly validated, translating safety-sensitive content inherently carries nuanced challenges that automated methods might not fully capture.

Suggestions for Improvement:
- Provide additional insights or experiments that quantify the impact of potential biases introduced by using a single evaluator

Conclusion:
Overall, this paper significantly advances multilingual safety evaluation for large language models, offering a robust methodology, insightful experimental results, and practical guidelines for future model improvements. Despite minor limitations primarily related to evaluator selection and translation challenges, the contribution is timely, valuable, and impactful for both academic researchers and industry practitioners.

---

### Decision · Program_Chairs · 2025-03-01

Accept